# Thyroid Hormone Regulates the Lipid Content of Muscle Fibers, Thus Affecting Physical Exercise Performance

**DOI:** 10.3390/ijms241512074

**Published:** 2023-07-28

**Authors:** Caterina Miro, Annarita Nappi, Serena Sagliocchi, Emery Di Cicco, Melania Murolo, Sepehr Torabinejad, Lucia Acampora, Arianna Pastore, Paolo Luciano, Evelina La Civita, Daniela Terracciano, Mariano Stornaiuolo, Monica Dentice, Annunziata Gaetana Cicatiello

**Affiliations:** 1Department of Clinical Medicine and Surgery, University of Naples “Federico II”, 80131 Naples, Italy; annarita.nappi@unina.it (A.N.); serena.sagliocchi@unina.it (S.S.); emery2304@gmail.com (E.D.C.); mel.murolo@gmail.com (M.M.); torabinejad.sepehr@yahoo.com (S.T.); acamporalucia7@gmail.com (L.A.); monica.dentice@unina.it (M.D.); annunziatagaetana.cicatiello2@unina.it (A.G.C.); 2Department of Pharmacy, University of Naples “Federico II”, 80149 Naples, Italy; arianna.pastore@unina.it (A.P.); pluciano@unina.it (P.L.); mariano.stornaiuolo@unina.it (M.S.); 3Department of Translational Medical Sciences, University of Naples “Federico II”, 80131 Naples, Italy; eva.lacivita@gmail.com (E.L.C.); daniela.terracciano@unina.it (D.T.); 4CEINGE–Biotecnologie Avanzate S.c.a.r.l., 80131 Naples, Italy

**Keywords:** thyroid hormone, fatty acids, skeletal muscle, exercise, lipids

## Abstract

Skeletal muscle (SkM) lipid composition plays an essential role in physiological muscle maintenance and exercise performance. Thyroid hormones (THs) regulate muscle formation and fuel energy utilization by modulating carbohydrates and lipid and protein metabolism. The best-known effects of THs in SkM include the promotion of mitochondrial biogenesis, the fiber-type switch from oxidative to glycolytic fibers, and enhanced angiogenesis. To assess the role of THs on the lipidic composition of SkM fibers, we performed lipidomic analyses of SkM cells and tissues, glucose tolerance experiments, and exercise performance tests. Our data demonstrated that TH treatment induces remodeling of the lipid profile and changes the proportion of fatty acids in SkM. In brief, THs significantly reduced the ratio of stearic/oleic acid in the muscle similar to what is induced by physical activity. The increased proportion of unsaturated fatty acids was linked to an improvement in insulin sensitivity and endurance exercise. These findings point to THs as critical endocrine factors affecting exercise performance and indicate that homeostatic maintenance of TH signals, by improving cell permeability and receptor stability at the cell membrane, is crucial for muscle physiology.

## 1. Introduction

Skeletal muscle (SkM) represents 40% of total body weight in humans and plays an essential role in whole-body energy expenditure. It is the primary site of lipid utilization, and it stands for the majority of whole-body insulin-mediated glucose uptake [1], both features closely connecting SkM to obesity and insulin resistance [2]. Insulin sensitivity is influenced by several membrane features, including the fatty acid lipid composition, the exposure of integral proteins (pivotally insulin receptor (IR) and glucose transporter 4 (GLUT-4)), and the expression of many different receptors for endocrine factors and signaling molecules, thereby regulating the overall insulin response [3]. Thus, a responsive asset of the sarcolemma in terms of fatty acid lipid composition is essential for total energy fueling and glucose metabolism, and its alteration may play a role in the pathogenesis of metabolic disorders, such as type 2 diabetes and obesity.

The membrane lipids composition is dynamic and constantly adapts to cholesterol and fatty acids (FAs) uptake and availability. The length and saturation of the membrane FAs, for example, impact the biochemical properties of the membrane, affecting the fluidity and permeability as well as the transport and functioning of its membrane proteins [4]. In cultured cells, increased mono- and poly-unsaturated fatty acids (MUFAs and PUFAs) concentration within cell membranes improves the membrane fluidity and the number of IRs [5]. The opposite effects have been observed when saturated FAs content in the membrane increases [6].

The regulation of the FAs composition of lipids, triglycerides, and phospholipids is under the control of environmental and endocrine factors influencing the FAs supply and turnover [7,8,9]. Several studies suggest that, among the physiological stimuli, exercise results in beneficial metabolic adaptations of SkM and leads to increased fat oxidation [10] and improved insulin sensitivity [11]. One of the mechanisms underpinning exercise effects on insulin sensitivity is SkM membrane remodeling [12]. Physical exercise potentiates glucose transport and elicits favorable changes in glucose and lipid metabolism, which is especially relevant to metabolic dysfunctions [13]. Moreover, exercise induces SkM growth traditionally referred to as SkM hypertrophy, which stands out as a rise in muscle mass, thickness, volume, and muscle fiber cross-sectional area (CSA) [14].

Thyroid hormones (THs) are potent regulators of whole-body energy expenditure and their fluctuation modulates glucose, lipid, and protein metabolism [15]. Indeed, altered TH status has profound clinical implications, and causes metabolic dysfunctions that affect body weight and energy expenditure [16]. In addition to the systemic TH supply by the thyroid gland, the tissue-specific modulation of THs by the deiodinase enzymes is critical in metabolic regulation [17]. By removing specific iodine atoms from its precursor thyroxine (T4), the deiodinases finely tune the availability of the active TH levels (T3) in target cells [18]. Type 2 deiodinase (D2) is the TH activator enzyme in SkM, acting as a central hub in muscle cell differentiation and physiology [19,20,21,22,23]. In addition to the modulation of fiber type shift and the role of THs in muscle cell differentiation, the metabolic pathways under TH control in SkM are still poorly recognized. Regarding lipid metabolism, THs activate both lipolysis and lipogenesis [24,25]. In particular, whether TH signaling can modulate lipids dynamic and, as a consequence, change muscle membrane FAs composition is not fully understood.

The aim of this study was to investigate whether THs influence the FAs composition of the SkM lipids and the relative effect on physical activity. In this study, we report that THs alter the proportion of saturated/unsaturated FAs. In detail, we observed a TH-mediated reduction in the stearic/oleic acids ratio in in vitro and in vivo muscle cell membranes. As a result, the muscle cell membrane composition is modified by THs, both in resting and post-exercise conditions. Interestingly, THs effects on membrane lipids composition were in part similar to those observed in the cell mechano-adaptation induced by the exercise. Moreover, the TH-induced lipid membrane remodeling was also associated with strong recruitment of the IR and GLUT-4.

## 2. Results

### 2.1. Thyroid Hormones Alter Lipids Profile and Saturated/Unsaturated FAs Ratio in Skeletal Muscle Cells

In a previous study, we metabolically profiled the differentially expressed metabolites in C2C12 muscle cells following TH treatment [21]. Interestingly, we observed that the hyperthyroid conditions enhanced the lipid precursor citrate and phosphatidylcholine (PC) and triacylglycerols (TAGs) levels in fluxomic experiments.

To get further insight into the impact of THs on the lipid profile of C2C12 cells, lipidomic analysis was carried out using high-resolution GC/MS. Cells were treated with 3 nM T3 and T4, which represent a mild hyperthyroid exposure, in order to avoid the effects of a supraphysiological condition. As shown in Figure 1A, we observed a significant decrease in the palmitic and stearic acids, belonging to the saturated FAs class (C16:0 and C18:0), and an increase in the corresponding MUFAs, palmitoleic and oleic acids (C16:1 and C18:1) induced by THs respect to control cells. The cholesterol levels did not significantly change. The Saturation Index (SI) of the cell membrane, indicated as the ratio of stearic acid (C18:0) to oleic acid (C18:1), and an indicator of membrane rigidity [26], was significantly reduced in TH-treated cells, suggesting that THs reduce the membrane rigidity (Figure 1B). Interestingly, apart from the altered ratio of saturated/unsaturated FAs, we did not observe any change in the total lipid content (Figure 1C).

Considering that variations in lipid fractions might be due to a general modulation of the carbon source metabolism, and also the main mode of THs action (namely the transcriptional regulation of TH-target genes), we measured the mRNA expression of key metabolic genes affecting glucose and lipid metabolism (Figure 1D–H). We found a shift from the PKM1-to-PKM2 isoform, which is preferentially expressed in fast muscle fibers and is indicated as a promoter of SkM hypertrophy [27] (Figure 1D,E). Moreover, we found a reduction in LDHA (suggested to specifically catalyze the pyruvate to lactate conversion) but not in LDH-B (the isoenzyme producing pyruvate from lactate) [28] (Figure 1D,F), and a reduction in PDHE1a that promotes the acetyl-CoA synthesis from pyruvate and is an index of carbohydrate oxidation [29] (Figure 1D). As for lipid metabolism-related genes, THs reduced the expression of lipolytic genes (ETFA and ACADM) (Figure 1G) and increased the expression of the Glycerol-3-Phosphate Acyltransferase 3 (GPAT3), the Sterol Regulatory Element Binding Transcription Factor 1 (SREBP) and the stearoyl-Coenzyme A desaturase 1 (SCD1), which are components of the lipogenesis pathway (Figure 1H). The elevation of these genes is in agreement with the increased proportion of unsaturated FAs [30]. Finally, we observed that TH treatment was associated with a slight increase in the thyroid hormone receptor α (TRα) expression (Figure 1I), while the thyroid hormone receptor β (TRβ) was only barely detected in our experimental conditions. These data suggest that exposure to low TH levels activates intracellular TH signaling and induces remodeling of membrane lipids composition associated with a higher unsaturated FAs fraction in hyperthyroid muscle cells.

### 2.2. Thyroid Hormones Reduce the Saturation Index of the Skeletal Muscle Membrane

To examine the in vivo impact of THs on the lipid profile of muscle fibers, we performed lipidomic analyses in gastrocnemius (GC) muscles isolated from euthyroid (resting EU) and mild hyperthyroid (resting Hyper) mice (Figure 2A). As expected, the THs treatment caused a significant increase in both systemic and intramuscular TH levels (Figure 2B,C). This condition of mild hyperthyroidism was associated with higher TRα and TRβ expression in GC muscles compared to euthyroid mice (Figure 2D). In line with the in vitro experiments, the analysis of FAs extracted from hyper- and eu-thyroid GC muscles showed a significant increase in the oleic acid (C18:1) and a reduction in the stearic acid (C18:0) (Figure 2E). Consequently, the hyperthyroid GC muscles showed a significant decrease in the SI compared to euthyroid muscles, 0.29 ± 0.08 vs. 1.25 ± 0.14, respectively (Figure 2F). As shown in Figure 2G, this decreased SI in hyperthyroid muscles was associated with high expression of GLUT-4, IR, and neutral lipids in muscle membranes, suggesting that the higher proportion of long-chain unsaturated FAs in the hyperthyroid muscles positively correlates with insulin sensitivity. These changes were not linked to relevant differences in muscle histology (Figure 2H). Hyperthyroid muscles also showed higher expression levels of glycolytic genes (Figure 2I,J) and a reduction in lipolytic compared to lipogenesis-related genes (Figure 2K,L). The determination of saturated and unsaturated FAs by GC-MS and the histological examination confirms that THs impact muscle membrane structure.

As expected, THs treatment induced also an MHCI-to-MHCII shift (Figure 3A–E); in detail, the mild hyperthyroid condition used in our experiments (see details in Section 4) determined an enrichment of MHC-IIa and IIx expression (Figure 3A–E).

### 2.3. The TH-Mediated Changes in Skeletal Muscle Membrane Lipidomic Profile Partially Reflect the Exercise-Induced Modifications

Exercise induces changes in SkM membranes [31,32], which result from the activation of specific signaling pathways, thereby influencing the transcription and translation of exercise-responsive genes. Since in hyperthyroid muscles we observed an alteration of membrane lipid composition, we asked if exercise induces the same changes, and also what role the TH-mediated membrane lipids modifications play in the functionality of SkM. In an attempt to address our issues, we first compared the lipidomic profiles of hyperthyroid muscles to those observed in exercise-trained muscles (Figure 2D and Figure 4A). Notably, we found a similar lipid pattern in the two different settings. In particular, we observed the same reduction in SI (stearic/oleic ratio) in exercise-trained muscles and in hyperthyroid muscles (Figure 2E and Figure 4B). Notably, the systemic TH levels were reduced by exercise, in spite of raised intramuscular TH concentrations (Figure 4C,D), in agreement with previous reports [33,34], highlighting the importance of measuring the intratissue modifications of THs besides the circulating levels. Importantly, the exercise stimulus also induced an increase in TRα and TRβ expression in GC muscles, suggesting an intramuscular activation of TH signaling (Figure 4E). These data were associated with an augmented localization of lipids in membranes and higher GLUT-4 and Insulin receptor β (IR) expression in exercise muscles similar to hyperthyroid muscles compared to euthyroid muscles (Figure 2F and Figure 4F,G). Moreover, the mRNA expression analysis of metabolic genes showed some similar changes induced by the two different stimuli, hyperthyroidism and exercise, suggesting that common and intersecting pathways are mediated by THs and physical exercise (Figure 2H–K and Figure 4H–K).

### 2.4. A Mild THs Stimulation Impacts the Structural and Functional Proprieties of Skeletal Muscle Influencing the Exercise Performance

In order to understand the impact of TH-mediated lipids membrane modifications on SkM functionality, we compared the exercise performance in hyper- and eu-thyroid mice [21]. In line with the dosage used in in vitro experiments and also to prevent the side effects of systemic hyperthyroidism, we tested two different doses of T3 and T4 in in vivo settings, referred to as mild hyperthyroidism (0.2 μg/mL T3 + 0.8 μg/mL T4 in drinking water) and high hyperthyroidism (1.0 μg/mL T3 + 4.0 μg/mL T4 in drinking water) (Figure 5A). These treatments resulted in correspondent T3 and T4 circulating levels, thus enabling us to establish an increasing grading of hyperthyroidism: euthyroidism, mild and high hyperthyroidism (Figure 5B). Interestingly, we observed an opposite regulation of TRs expression profile in the different hyperthyroid conditions (Figure 5C). In detail, the mild hyperthyroidism increased the TRs expression, while the high dose treatment was instead associated with a reduction in TH receptors in exercised muscles. We evaluated both the resistance and high-speed power training. When the mice were trained at the same moderate speed protocol, mild hyperthyroidism rendered the mice more resistant, as measured by longer running time and distance, and a lower number of interruptions per minute (LB/min), without any effect on muscle power compared to euthyroid mice (Figure 5D–G). However, when the mice were subjected to an increasing speed protocol, the mild hyperthyroid mice reached a lower maximal speed compared to the euthyroid mice (0.24 m/s vs. 0.28 m/s, respectively) (Figure 5H). Interestingly, as expected, supra-physiological hyperthyroidism negatively impacted on muscle performance, most likely due to the well-known side effects of THs on cardiopulmonary activity (Figure 5D–H). These differences in exercise ability were associated with enhanced expression of the mitochondrial biogenic factor PGC-1α in mild-TH-treated and exercised mice, and this effect was further increased in mild-TH-treated and exercised mice (Figure 5I).

The TH-induced reportioning of saturated and unsaturated FAs strongly impacted muscle histology with a higher localization of lipids in membrane and the expression of insulin-dependent glucose metabolism factors GLUT4 and IR (Figure 6A–D). To further investigate if exercise affects only the synthesis or also the translocation of GLUT-4, we performed a protein fractioning method allowing the isolation of heavy membranes (HM, representing the plasma membranes, mitochondria, and RER), light membranes (LM, representing the SRE, vesicles and free polysomes) and cytosol. Notably, we observed that while THs and exercise led to a global increase in GLUT-4 (in total homogenates), THs and exercise alone increased mostly the LM fraction, while the combined effect of exercise and THs enhanced the GLUT-4 in HM, with a critical implication for insulin sensitivity (Figure 6E,F). Finally, the mRNA expression analysis of metabolic genes confirmed a TH-mediated metabolic rearrangement of muscle fibers in order to meet the higher energy demand in response to exercise (Figure 6G–I).

### 2.5. Systemic Hyperthyroidism Improves Insulin Sensitivity In Vivo

While insulin resistance has been correlated with changes in the Saturation Index of phospholipids in muscle [35,36], the direct cause has not been proven. Since we observed that IR and GLUT-4 are elevated by mild THs treatment in SkM, we investigated the overall effects of THs on insulin sensitivity by performing glucose and insulin tolerance tests (GTT and ITT assays) and collected tissues for phospho-Akt and p-IR western blots analysis. While GTT showed that high THs result in even higher blood glucose levels compared to euthyroid mice, probably due to the sustained gluconeogenesis [37] (Figure 7A,B), ITT demonstrated that high THs ameliorate the ability of insulin to reduce blood glucose, thus arguing for a permissive effect of THs on insulin action (Figure 7C,D). This phenotype was associated with the activation of the insulin pathway in hyperthyroid GC muscles, and this effect was also potentiated in combination with insulin both in vivo and in vitro (Figure 7E,F). The analysis of GLUT-4 protein levels in different muscle types (GC, soleus, and EDL muscles) revealed that THs sustain GLUT-4 expression in different muscles regardless of fiber composition (Figure 7G).

Altogether, our results demonstrate that thyroid status influences the FAs composition of the SkM lipids, affecting membrane fluidity and in turn insulin sensitivity, glucose homeostasis, and physical activity.

## 3. Discussion

Skeletal muscle is a plastic tissue, able to rapidly adapt its structural components in response to external stimuli [31]. The extreme adaptability of SkM to environmental changes is due to its high structural and metabolic flexibility. Indeed, SkM can rapidly adjust the rate of ATP synthesis, perfusion, and substrate utilization, depending on energy needs [38].

Physical exercise is the major stimulus inducing SkM adaptations by changing the metabolic potential, morphology, and physiology of myofibers [39]. In turn, these adaptations improve SkM performance [40]. Therefore, regular physical activity is ameliorative for health and is highly recommended for the prevention of several chronic pathological conditions. Noteworthy, an alteration of TH levels in extreme conditions of hypo- and hyperthyroidism has deleterious effects. However, a critical issue is that moderate THs fluctuations can be beneficial, depending on the delicate balance of THs-regulated processes. Hence, understanding the role of THs in exercise-induced molecular events leading to SkM remodeling is a challenging issue. Notably, it is well known that THs are the upstream regulators of the transcription of several genes related to myogenesis (MyoD, Myogenin, MHC isoforms), contractility (SERCA, Troponin, MHC), and energetics (PGC-1α, Na/K-ATPAse, UCP3, GPT2, etc.) [16,21,31,41]. Moreover, TH receptors TRα and TRβ are both critical regulators of muscular genes and alterations in TRs are associated with muscle dysfunctions in mice and humans [24,42,43]. In addition to the genomic regulation of myogenic markers, the non-genomic action of TH also contributes to muscle fiber differentiation and metabolism [44,45].

In our study, we found that THs alter the proportion of saturated and unsaturated FAs in SkM lipids membrane. In particular, we measured a higher oleic acid concentration in TH-treated muscle cells and tissues. We observed that the changes in oleic acid membrane levels were associated with modulations in the function of proteins related to glucose metabolism. We found higher recruitment of IR and GLUT-4 in muscle membranes following THs treatment, two factors converging in higher insulin sensitivity.

Insulin sensitivity is also correlated to the degree of physical activity, and exercise has been shown to improve insulin action in insulin-resistant subjects [46]. This effect is mediated by the improvement of the molecular abnormalities that are responsible for insulin resistance, contributing in this way to restoring physiological insulin sensitivity. Indeed, individuals with diabetes due to insulin resistance can be reversed by habitual physical exercise [47]. It is being established that exercise-induced benefits may be augmented by appropriate dietary manipulations [48]. In this perspective, molecular research aims to identify key signaling pathways and proteins involved in the changes in insulin sensitivity during exercise. Here, we argue that THs are among the beneficial factors for the SkM membrane plasticity and that THs contribute to the improvement of insulin signaling. By enriching the muscle membrane of the oleic acid, THs confer the membrane elasticity essential for the modulation of membrane proteins activity and translocation and dictate structural and metabolic fingerprinting, thus impacting muscle performance. Indeed, besides the re-proportioning of stearic-to-oleic acid content, a mild THs treatment induced a metabolic reprogramming of muscle fibers, with enhanced expression of PKM2 over the PKM1 isoform that is also typical of trained MHC-IIx rich fast fibers [27], and an increase in the LDHA over the LDHB isoform associated with OxPhos-to-glycolisis shift [28]. Notably, the induction of PKM2 and LDHA is moreover a hallmark of the Warburg effect in cancer cells, which is also induced by the elevation of TH signal in cancer [49,50,51,52].

Our work describes a key role for THs in SkM physiology that is strictly in agreement with previously described functions of THs in SkM. Indeed, it has been widely reported that THs contribute to mitochondrial biogenesis through the direct induction of PGC-1α, which is also potently induced by physical exercise, thus reinforcing the interplay between TH action and SkM activity [53]. Moreover, in our experimental settings, we observed that THs induce a shift from MHCI to MHCII fibers, and in particular an increase in MHC-IIa and -IIx proteins, which are typical of intermediate muscle fibers, with a mixed oxidative/glycolytic metabolism [31].

Moreover, by testing a different range of TH treatments (mild and high hyperthyroidism), our work provides new evidence of the differential physiological response to varying doses of THs. The analysis of TRα and TRβ expression showed that mild hyperthyroidism was associated with an increase in TRs expression in both resting and exercised muscles, while the high-dose treatment was instead associated with a reduction in TH receptors. The improvement in exercise performance that we observed in the mild hyperthyroidism condition could be attributed to an activation of intracellular TH signaling through the elevation of TR expression. In contrast, the detrimental effect on exercise performance in mice exposed to excess levels of TH was associated with the suppression of TR expression. The changes in TRs expression in different hyperthyroid status suggest that skeletal muscle is able to modulate its intracellular TH action. The downregulation of TRs might represent a compensatory mechanism to prevent tissue-level alterations in response to strong systemic hyperthyroidism.

Strikingly, here, we observed that the metabolic changes induced by a mild hyperthyroid condition mirror the exercise-induced adaptations of skeletal muscle. In detail, we observed that both THs treatment and exercise increase the expression of GLUT-4 in myofibers and that combined exercise in mild hyperthyroid mice leads to a higher translocation of GLUT-4 in the sarcolemma. The connections between the muscle molecular adaptations in exercised animals and in mild hyperthyroid mice leads us to question whether exercise can elevate TH signaling. Importantly, we observed that mice post-exercise undergo a reduction in systemic THs concentration in spite of increased intramuscular TH levels.

Our results support a model in which THs promote muscle membrane plasticity through the elevation of oleic acid proportion, providing a mechanism that might explain, at least in part, some of the protective effects of THs against dyslipidemias and insulin resistance. Nevertheless, it is worth mentioning a limitation of our study using a systemic hyperthyroid treatment to study intramuscular adaptation, which does not allow avoiding that systemic hyperthyroidism induces extra-muscular modification, in turn affecting skeletal muscle.

Similarly, considering the pleiotropic functions of THs, especially in cardiac muscle (i.e., hyperthyroidism is known to increase heart rate, pulse pressure, and cardiac output in humans), we cannot exclude that part of TH-induced improvement in exercise performance may be due to enhanced cardiac output. These complex interactions between different organs can make it challenging to determine the specific mechanism for improved exercise performance and suggests that future studies are needed to address the contribution of locally modified TH levels to exercise performance.

The present findings have the potential to be relevant for muscle physiology, as well as for our understanding of the complex interplay between endocrine and mechanical signals orchestrating muscle contraction.

## 4. Materials and Methods

### 4.1. Cell Lines and Culture Conditions

C2C12 cells were obtained from ATCC and cultured in DMEM High Glucose (HiMedia Leading BioSciences Company, Mumbai, Maharashtra, India, cod. AL007) supplemented with 10% FBS (HiMedia Leading BioSciences Company, cod. RM10432), 2 mM glutamine (Gibco, Thermo Fisher Scientific, Waltham, MA, USA, cod. 25030024), 50 i.u. penicillin, and 50 μg/mL streptomycin (Gibco, Thermo Fisher Scientific, cod. 15070063). Cells are grown at 37 °C in a humidified incubator with 5% CO_2_. For lipidomic analyses, C2C12 cells were grown at 40–50% confluence and treated with THs (T3, and T4, Sigma-Aldrich, St. Louis, MI, USA, cod. T6397 and cod. T2501, respectively) at 3.0 nM final concentration each one for 48 h.

### 4.2. Mouse Strains

Three-month-old male C57BL/6J mice were acquired from Jackson Laboratory (Bar Harbor, ME, USA) and used in this work. Animal research was performed according to national and European community guidelines [54,55]. Mice were fed regular chow *ad libitum* and maintained in fresh bedding cages under controlled conditions, namely 20–24 °C, 50–60% humidity, and a 14:10 light–dark cycle. All in vivo procedures received the approval of the Institutional Animal Care and Use Committee (IACUC) (protocol n. 354/2019-PR).

### 4.3. Elisa Immunoassay

Blood samples were collected in 1.5 mL tubes without anticoagulant. After clot formation, the samples were centrifuged, and the serum was recovered and stored at −20 °C until assayed. Total T4 and total T3 were measured by the ADVIA Centaur XP Immunoassay system using a commercial kit, as recommended by the manufacturer (Siemens Healthcare Diagnostics, Camberley, UK).

### 4.4. LC-MS/MS Analysis for Measurements of Intramuscular Thyroid Hormones

The intramuscular T3 and T4 levels were measured as previously described [56,57]. Briefly, frozen tissue samples were homogenized on ice in phosphate buffer containing 0.25 M sucrose, 1 mM EDTA, 0.1 M NaPO4, and 10 mM DTT, and then sonicated. Then, 100 µg of protein extract was incubated for 1 h at 37 °C in 0.3 mL PE buffer with 10 mM DTT. Then, 20 nM T4-13C6 was added to the reactions and incubated at 37 °C for 6 h. Proteins were precipitated in acetonitrile and supernatant evaporated to dryness at 37 °C in a rotavapor. The dried extract was reconstituted with 30 µL of a 95:5 methanol:NH4-OH solution and centrifuged at 13,000 rpm for 10 min. Samples (5 µL) were injected on a Raptor Biphenyl 2.7 µm, 100 mm × 2.1 mm (cat. 9309A12) using as Mobile Phase A 0.1% Formic acid in water and as Mobile phase B 0.1% Formic acid in methanol. Calibration curves (1 to 40 nM) were prepared dissolving pure T4, T3, and T4-13C6 in 95:5 methanol: NH4-OH solution. The analyte MRMs were for T3-13C6 (658.07 > 612.1 > 514.1); T3 (652.07 > 606.1 > 508.1); T4 (778.0 > 731.9 > 323.9); T4-13C6 (784.1 > 738.04). Measurement was conducted in triplicate.

### 4.5. Treadmill Exercise Running

Twelve-week-old male C57BL/6 mice were randomly assigned to one of four experimental groups of nine mice per group: the euthyroid sedentary control (resting EU), the hyperthyroid sedentary (resting Hyper), the euthyroid exercise (exercise EU), or the hyperthyroid exercise (exercise Hyper) group. The hyperthyroid group received THs in drinking water at two different doses: mild hyperthyroidism (0.2 mg/mL T3 + 0.8 mg/mL T4) and high hyperthyroidism (1.0 mg/mL T3 + 4.0 mg/mL T4) [21]. At the beginning of TH supplementation, the mice were single housed to avoid competition in water intake. Mice assigned to exercise were submitted to the increasing speed protocol as previously described [58] to test the maximal speed and set the exercise intensity before the endurance exercise protocol. The endurance exercise protocol consists of treadmill running at 60% of maximal speed until exhaustion. Mouse exhaustion is defined as 10 continuous seconds of no longer attempting to run on the belt. All tests and exercise protocols were carried out with a motor-driven treadmill (TSE Systems, Berlin, Germany). Animals undergoing exercise were euthanized 24 h after the last exercise session to isolate the effects of acute exercise response. Sedentary mice were housed in cages without any stimulus to movement. At the experimental endpoint, the blood, gastrocnemius (GC) and tibialis anterior (TA) muscles were collected and stored at −80 °C for molecular, histological, and lipidomic analysis.

### 4.6. Lipidomic Analysis

Lipidomic analysis was performed starting from muscle tissues (50 mg) or C2C12 cells (2 × 10^6^ cells) by GC-MS as previously described [59]. For TMS derivatization, the reactions were carried out in 180 min using samples dissolved in pyridine (50 µL) and derivatized with 25 µL of N,O-Bis(trimethylsilyl(TMS)trifluoroacetamide (BSTFA). For FAME derivatization, samples were resuspended in hexane (3 mL), trans-esterified at room temperature with methanolic KOH 2 N (0.5 mL) and then quenched after 3 min with water (3 mL). Upper organic phases were dried and then resuspended in 100 µL of hexane. For both TMS and FAME derivatized samples, one µL was injected, establishing a 1:10 split ratio. GC-MS analyses were performed using a Shimadzu GCMS 2010plus (Kyoto, Japan) with well-settled parameters as previously described [60]. Five replicates of each injection were carried out. MS spectra were elaborated with Compass Data Analysis version 4.2 (Bruker) and the identification of compounds was based on a comparison of NIST library standards. Metabolite signals were normalized using internal standards.

For GC-MS analyses, dried membrane pellets were dissolved in 1 mL of ice-cold dichloromethane. Insoluble material was removed by centrifugation at high speed for 10 min at 4 °C. The supernatants were dried and resuspended in hexane. Fatty acid methyl esters of total cellular lipids were prepared by transmethylation using hexane and methanolic 2M-KOH. Samples were thus injected, and split ratio 1:10 and GC-MS analyses were carried out on a Shimadzu GCMS 2010plus (Kyoto, Japan) with the following parameters: injector and detector temperature was 250 °C; the oven temperature was 180 °C, held for 6 min then increased to 250 °C at 3 °C/min; the final temperature was held for 10 min. The carrier gas (He) flow rate was 1.0 mL/min. FAME identity was confirmed using an internal library and quantitated with calibration curves.

### 4.7. Glucose Tolerance Test and Insulin Tolerance Test

A glucose tolerance test (GTT) was performed after overnight fasting. Blood collection was performed at time 0, then mice were injected intraperitoneally (ip) with a 20% glucose solution (2.0 g/kg body weight). Blood samples were collected at 15, 30, 60, 90 and 120 min after glucose injection for determination of glycemia. An insulin tolerance test (ITT) was performed after 4 h of fasting. Blood samples were collected before or 15, 30, 60, 90, and 120 min after ip insulin injection (1.0 U/kg) for serum glucose determination. Blood glucose (mg/dL) was measured using CountourXT (Bayer, Leverkusen, Germany).

### 4.8. Quantitative RT-PCR

RNA preparation and quantitative (q)RT-PCR were performed as previously described [21]. In brief, messenger RNAs were extracted with Trizol reagent (Life Technologies) starting from 6 × 10^6^ C2C12 cells or 100 mg GC muscles. Complementary DNAs were prepared with SuperScript VILO Master Mix (Life Technologies, Carlsbad, CA, USA) as indicated by the manufacturer starting from 1000 ng of total RNA (A 260/280 ratio of ~1.8–2.0). The cDNAs were amplified by PCR in a CFX Connect Real-Time PCR Detection System (Bio-Rad, Hercules, CA, USA) with the fluorescent double-stranded DNA-binding dye SYBR Green (BioRad). Specific primers for each gene were designed to work under the same cycling conditions (95 °C for 10 min followed by 40 cycles at 95 °C for 15 s and 60 °C for 1 min), thereby generating products of comparable sizes (about 200 bp for each amplicon). Primer combinations were positioned whenever possible to span an exon–exon junction and the RNA digested with DNase to avoid genomic DNA interference. The relative amounts of gene expression were calculated using Cyclophilin-A as an internal standard. All samples were run in triplicate. The results, expressed as N-fold differences in target gene expression, were determined as follows = 2^−(DCt target − DCt control)^ [61]. Table 1 shows the mouse-specific primer pairs used in qRT-PCR experiments.

### 4.9. Western Blot Analysis

Freeze-dried GC muscles were pulverized using a mortar and pestle chilled with liquid nitrogen. Then, 50 mg of samples were homogenized in 0.2 mL of lysis buffer with a protease inhibitor cocktail (Sigma-Aldrich) with the addition of activated sodium orthovanadate (Sigma-Aldrich, St. Louis, MI, USA). The homogenate was subsequently centrifuged at 3000× *g* at 4 °C for 30 min to remove the tissue debris. The protein concentration of the supernatant was then quantified using a Bio-Rad DC protein assay using BSA as a standard per the manufacturer protocol (Bio-Rad, Hercules, CA, USA). The protein extract was then boiled for 7 min with 1:1 volume of 2× Laemmeli loading buffer. Thirty micrograms of protein from each sample were then separated using 10% SDS-PAGE gel and transferred onto an Immobilon-P transfer membrane (Millipore, Burlington, MA, USA). The membrane was blocked with 5% non-fat dry PBS milk, and consecutively probed with an anti-GLUT-4 primary antibody (anti-Glucose Transporter GLUT-4 [1F8] ab35826, Abcam, Cambridge, UK), and secondary antibody (anti-mouse IgG-HRP BioRad, Hercules, CA, USA, cod. 1706516, 1:3000). The signal was detected by chemiluminescence (Millipore, Burlington, MA, USA, Cat. No. WBKLS0500). After extensive washing, the membrane was incubated with an anti-GAPDH antibody (Elabscience, Houston, TX, USA, cod. E-AB-20059) as the loading control. Western Blots were run in triplicate, and bands were quantified with ImageJ 1.52k Wayne Rasband, NIH (Bethesda, MD, USA) (http://imagej.nih.gov/ij) (Table 2).

### 4.10. Cell Fractionation

Cell fractionation and analysis from gastrocnemius muscle were performed as described previously [62]. Briefly, 100 mg of gastrocnemius muscle was resuspended in 0.5 mL of ice-cold buffer B (150.0 mM NaCl, 10.0 mM TRIS HCl pH 7.4, 1.0 mM EDTA, supplemented with protease inhibitors), passed through a 21-gauge needle 20 times. A 50 µL quantity of the resuspended sample was stored by transferring it into a new 1.5 mL microcentrifuge tube and used as total homogenate. The remaining fraction was centrifuged at 400 rpm for 5 min at 4 °C. The resulting supernatant was centrifuged at 14,000 rpm for 10 min at 4 °C in an Eppendorf centrifuge to pellet the Heavy Membranes (HM). The supernatant was centrifuged at 45,000 rpm for 30 min at 4 °C in a Beckman Coulter TL-100 centrifuge (Brea, CA, USA) to pellet the Light Membranes (LM) and isolate the Cytosolic fraction. The collected fractions were resolved on linear 10% polyacrylamide gel and analyzed for the expression of anti-GLUT-4 primary antibody (anti-Glucose Transporter GLUT-4 [1F8], ab35826, Abcam), anti-CNX (anti-Calnexin, ADI-SPA-860-F, Enzo Life Sciences, Pero (MI) Italy) and anti-GAPDH antibody (Elabscience, cod. E-AB-20059). The intensities of the corresponding bands were determined and quantified with ImageJ 1.52k Wayne Rasband, NIH (Bethesda, MD, USA) (http://imagej.nih.gov/ij).

### 4.11. Histology, MHC Immunostaining, and BODIPY (493/503) Staining and Visualization

Hematoxylin/Eosin (H&E) staining (Sigma-Aldrich, St. Louis, MO, USA) was carried out on 8 μm muscle cryosections according to classical methods [63,64]. Fiber size distribution was measured by considering six hundred myofibers per muscle using ImageJ 1.52k Wayne Rasband, NIH (Bethesda, MD, USA) (http://imagej.nih.gov/ij). For IR, GLUT-4, and MHC immunostaining, sections were fixed with 4% paraformaldehyde or 100% methanol for 15 min, blocked with 3% normal goat serum and 0.3% Triton X-100 in PBS and sequentially incubated with the primary and secondary fluorescent antibodies (Table 2). Then, nuclei were DAPI stained and mounted in 80% glycerol. The quantification of myofiber type was conducted by quantifying the total fluorescent signal expressed as integrated density and by counting the number of positive fibers compared to the total fibers in the same field of view [65].

To visualize the neutral lipid content, we used the fluorescent neutral lipid dye 4,4-difluoro-1,3,5,7,8-pentamethyl-4-bora-3a,4a-diaza-s-indacene (BODIPY 493/503). Bodipy (D3922, Molecular Probes, Eugene, OR, USA) (excitation wavelength 480 nm, emission maximum 515 nm) was diluted in PBS at a concentration of 1 mg/mL and applied to cross sections of muscle fibers for 30 min. We assessed fixed (4% paraformaldehyde for 5 min) sections; 4,6-diamidino-2-phenylindole (DAPI) was used to identify nuclei. Following fixation, samples were washed 3 times in phosphate-buffered saline (PBS) for 10 min. All samples were mounted in 80% glycerol and covered with glass coverslips (No. 1, VWR). Images were captured with a fluorescent Leica DMi8 microscope using the Leica Application Suite LAS X Imaging (Leica, Berlin, Germany) for the acquisitions.

### 4.12. Statistical Analysis

The results are represented as the mean ± Standard Deviation (SD) throughout. Differences between samples were assessed by the Student’s two-tailed *t*-test for independent samples. Relative mRNA levels (in which the control sample was arbitrarily set as 1) are reported as results of Real-Time qRT-PCR, in which the expression of cyclophilin-A served as the housekeeping gene.

Multiple comparisons and differences were analyzed for statistical significance by the two-way ANOVA test and Bonferroni posttest analysis [66]. All graphs, bars, or lines indicate the mean, and error bars indicate the standard error of the mean (SEM).

In all experiments, differences were considered significant when the *p*-value (*p*) was less than 0.05. Asterisks indicate significance at * *p* < 0.05, ** *p* < 0.01, and *** *p* < 0.001 throughout. Definitions of *n*-values are reported in figure legends.

## 5. Conclusions

In conclusion, our study shows that THs are important endocrine factors regulating the lipid composition of SkM and that the metabolic changes imposed by a “low dose” treatment with THs positively correlate with insulin sensitivity and membrane plasticity. In turn, this reinforces the concept that homeostatic maintenance of a functional TH status is a strict demand to sustain muscle performance.

## Figures and Tables

**Figure 1 ijms-24-12074-f001:**
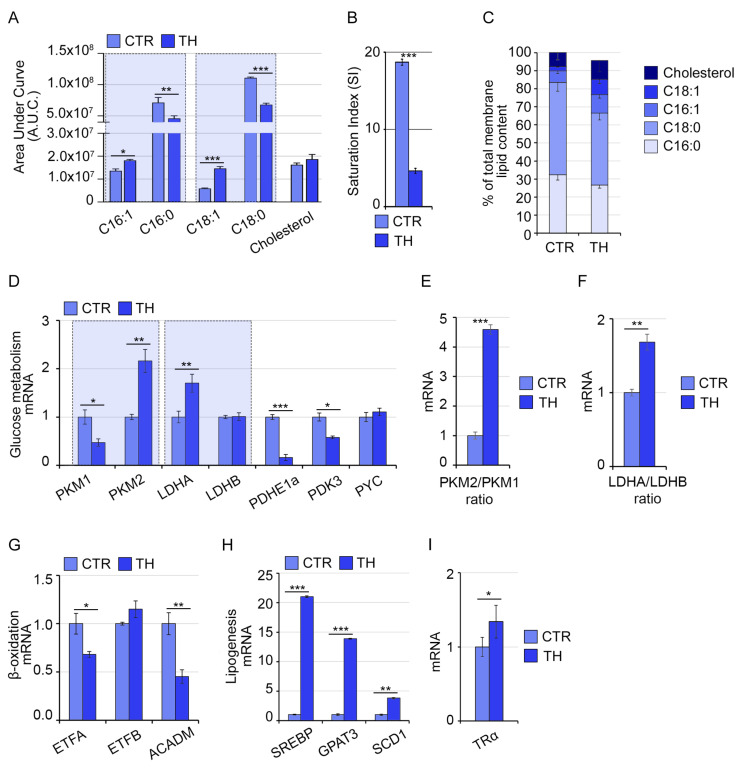
THs treatment changes the proportion of oleic/stearic acids in C2C12 skeletal muscle cells. (**A**) Lipidomic analysis performed by GC-MS of membrane lipids in C2C12 skeletal muscle cells altered by THs treatment. Metabolites signals were normalized using internal standards and expressed as Area Under Curve (AUC). The two-way Anova test and Bonferroni post-test were exploited to analyze statistical significance. (**B**) Saturation Index (SI) levels (stearic/oleic acid ratio) in C2C12 muscle membranes from controls and THs-treated cells. All graphs indicate the mean and error bars indicate Standard Error of the Mean (SEM). Values are mean ± SEM (*n* = 5). * *p* < 0.05, ** *p* < 0.01, *** *p* < 0.001 (CTR vs. TH). (**C**) Total lipid content measured by GC-MS lipidomic analysis indicating the relative percentage of palmitic and stearic saturated FAs (16:0 and 18:0), of palmitoleic and oleic unsaturated FAs (16:1 and 18:1) and cholesterol respect to total in C2C12 THs-treated cells and control cells. (**D**) mRNA expression analysis of indicated genes involved in glycolytic pathway in C2C12 THs-treated cells compared to control cells. (**E**) Ratio of PKM2/PKM1 mRNA expression levels in same cells as in (**D**). (**F**) Ratio of LDHA/LDHB mRNA expression levels in same cells as in (**D**). (**G**,**H**) mRNA expression analysis of indicated genes involved in lipolytic (**G**) and lipogenesis (**H**) pathways in same cells as in (**D**). (**I**) mRNA expression levels of TRα in same cells as in (**D**). Cyclophilin-A was used as an internal control. Normalized copies of the genes in control cells were set as 1. Data represent the mean ± SD of nine replicates. * *p* < 0.05, ** *p* < 0.01, *** *p* < 0.001.

**Figure 2 ijms-24-12074-f002:**
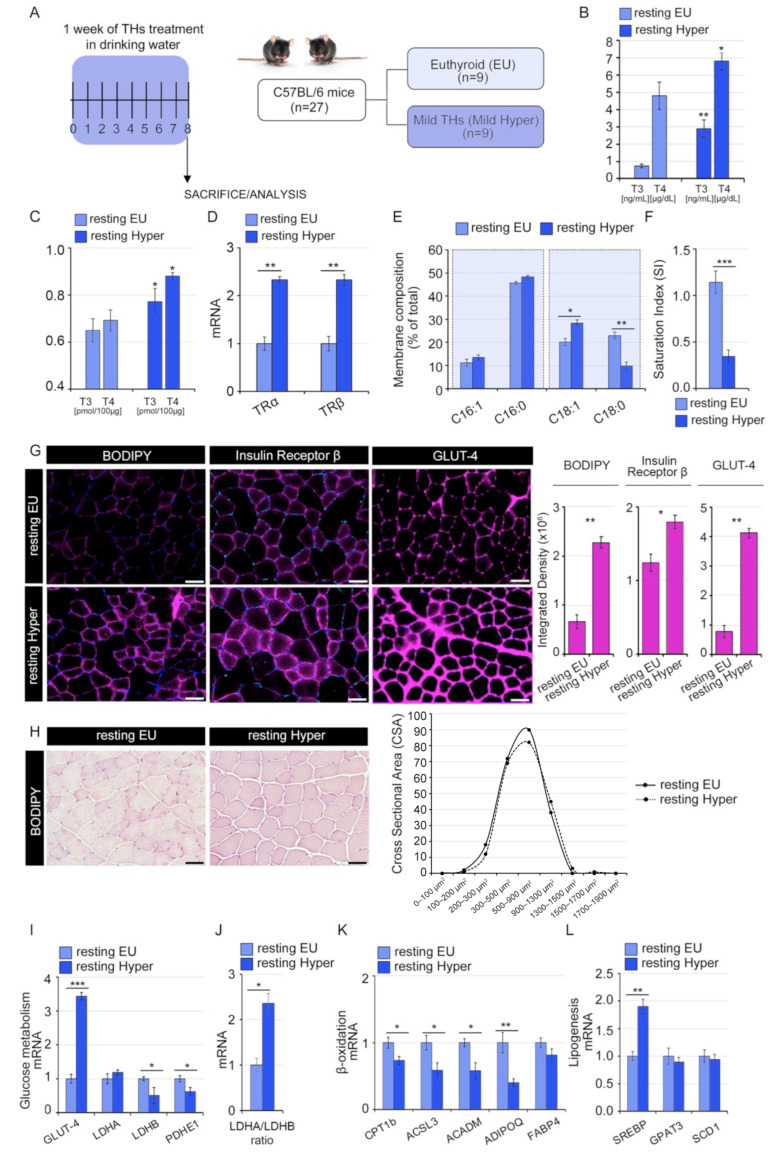
Mild hyperthyroidism impacts on the membrane lipid profile of skeletal muscle tissues. (**A**) Schematic representation of experimental setting and resting animal groups: euthyroid and mild hyperthyroid mice (resting EU vs. resting Hyper). (**B**) Serum levels of T3 (ng/mL) and T4 (µg/dL) in resting euthyroid and hyperthyroid mice measured by immunoassay. The statistical significance was calculated compared to the resting EU. Data represent the mean ± SD of five replicates. * *p* < 0.05, ** *p* < 0.01. (**C**) Intramuscular levels of T3 (pmol/100 µg) and T4 (pmol/100 µg) in resting euthyroid and hyperthyroid mice measured by LC-MS. The statistical significance was calculated compared to the resting EU. Data represent the mean ± SD of five replicates. * *p* < 0.05. (**D**) mRNA expression levels of TRα and TRb in resting hyper compared to EU GC muscles. Cyclophilin-A was used as an internal control. Normalized copies of TR genes in resting EU GC muscles were set as 1. (**E**) Lipidomic analysis performed by GC-MS of SkM membrane lipids extracted from gastrocnemius (GC) muscles of mild hyperthyroidism compared to euthyroid mice. Metabolites signals were normalized using internal standards and expressed as Area Under Curve (AUC). Comparisons and differences were analyzed for statistical significance by two-way Anova test and Bonferroni posttest analysis. (**F**) SI levels (stearic/oleic acid ratio) in same GC muscles as in (**C**). All graphs indicate mean and error bars indicate Standard Error of the Mean (SEM). Values are mean ± SEM (*n* = 9). * *p* < 0.05, ** *p* < 0.01, *** *p* < 0.001. (EU vs. Hyper). (**G**) Immunofluorescence analysis of Bodipy for lipid staining, Insulin Receptor β (IR β), and GLUT-4 in Tibial Anterior (TA) muscles as in (C). Magnification 20×. Scale bar, 50 μm. (**H**) H&E analysis of TA as in (C). Magnification 20×. Scale bar, 50 μm. The graphs below represent the Cross-Sectional Area (CSA) of muscle fibers. (**I**) mRNA expression levels of genes involved in glycolytic pathway in EU and Hyper GC muscles. (**J**) Ratio of LDHA/LDHB mRNA expression levels as in (**G**). (**K**,**L**) mRNA expression analysis of lipolytic and lipogenesis-related genes as in (**G**). Cyclophilin-A was used as internal control. Normalized copies of the indicated genes in EU GC muscles were set as 1. Data represent the mean ± SD of nine replicates. * *p* < 0.05, ** *p* < 0.01, *** *p* < 0.001.

**Figure 3 ijms-24-12074-f003:**
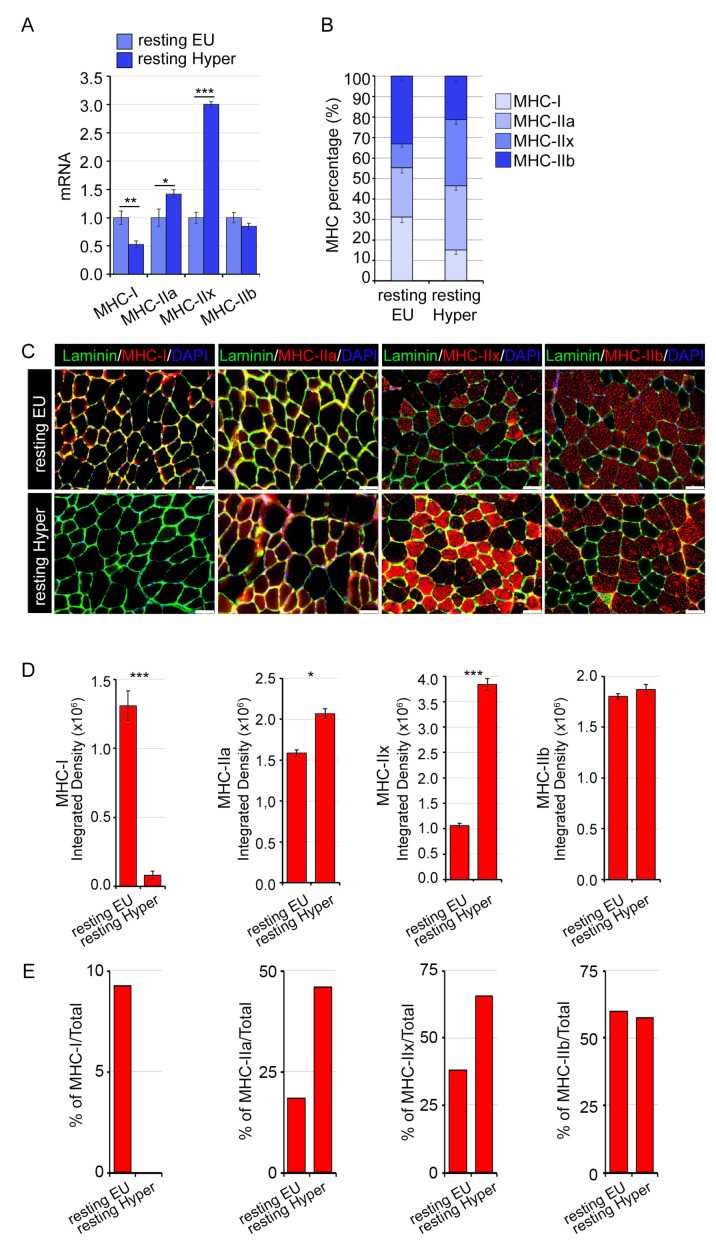
Mild hyperthyroidism induces an MHC-I to MHC-II muscle fibers with a higher MHC-IIa and MHC-IIx proportion. (**A**) mRNA expression levels of MHC-I, MHC-IIa, MHC-IIx, and MHC-IIb in hyper- and eu-thyroid GC muscles (resting EU and resting Hyper). (**B**) Relative percentage of MHC-I and MHC-II mRNA levels respect to total of MHC in hyperthyroid compared to euthyroid GC muscles. Levels of indicated genes are relative to Cyclophilin-A mRNA used as an internal control and normalized to the euthyroid muscles. (**C**) Immunofluorescence analysis of MHC-I, MHC-IIa, MHC-IIx, and MHC-IIb in hyper- and euthyroid TA muscles (resting EU and resting Hyper). Magnification 20×. Scale bar, 50 μm. (**D**) Relative quantification of the mean integrated density of the fluorescence signal in (**C**). (**E**) Quantification is calculated by counting the number of positive fibers compared to the total fibers in the same field of view of the fluorescence staining in (**C**). Data represent the mean ± SD of nine replicates. * *p* < 0.05, ** *p* < 0.01, *** *p* < 0.001.

**Figure 4 ijms-24-12074-f004:**
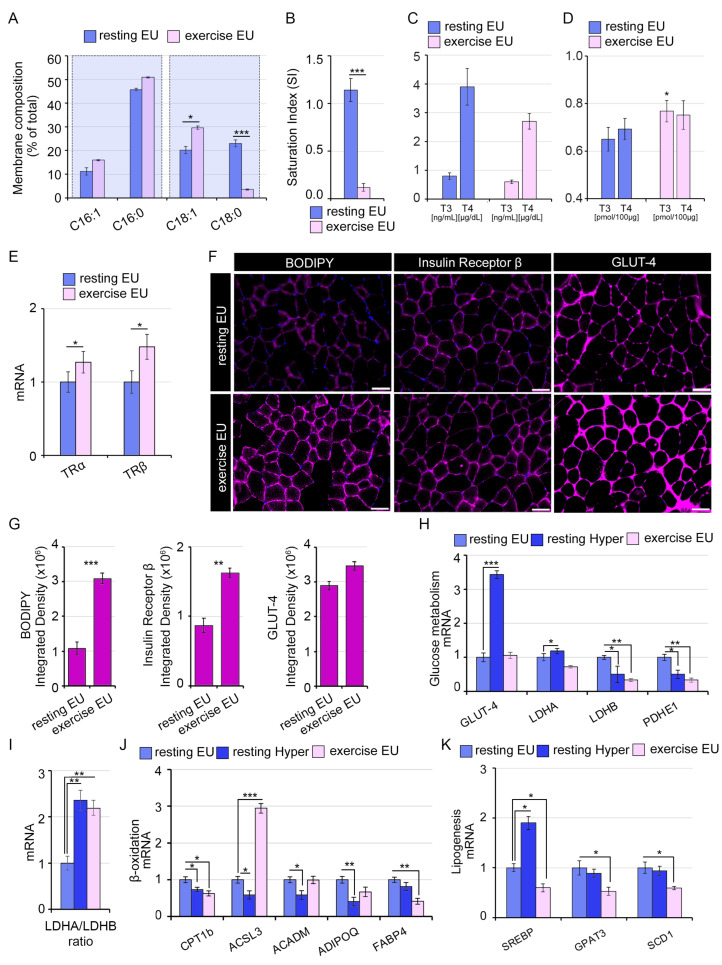
THs partially reflect the exercise-induced modifications of SkM membrane’s lipidomic profile. (**A**) Lipidomic analysis performed by GC-MS of SkM membrane lipids extracted from GC muscles of exercise-trained compared to resting euthyroid mice. Metabolites signals were normalized using internal standards and expressed as Area Under Curve (AUC). Comparisons and differences were analyzed for statistical significance by two-way Anova test and Bonferroni posttest analysis. (**B**) SI levels (stearic/oleic acid ratio) in same GC muscles as in A. All graphs indicate the mean and error bars indicate the Standard Error of the Mean (SEM). Values are mean ± SEM (*n* = 9). * *p* < 0.05, *** *p* < 0.001 (resting EU vs. exercise EU). (**C**) Serum levels of T3 (ng/mL) and T4 (µg/dL) in resting and exercised mice measured by immunoassay. (**D**) Intramuscular levels of T3 (pmol/100 µg) and T4 (pmol/100 µg) in resting and exercised GC muscles measured by LC-MS. The statistical significance was calculated compared to resting EU. Data represent the mean ± SD of five replicates. * *p* < 0.05. (**E**) mRNA expression levels of TRα and TRβ in exercised EU compared to resting EU GC muscles. Cyclophilin-A was used as internal control. Normalized copies of TR genes in resting EU GC muscles were set as 1. (**F**) Immunofluorescence analysis of Bodipy for lipid staining, Insulin receptor β, and GLUT-4 in TA muscles as in A. Magnification 20×. Scale bar, 50 μm. (**G**) Histograms indicate the quantification of fluorescence signal expressed as the integrated density of the immunofluorescences shown in (**F**). (**H**) mRNA expression levels of genes involved in glycolytic pathway in exercise EU, resting Hyper compared to resting EU GC muscles. (**I**) Ratio of LDHA/LDHB mRNA expression levels as in (**H**). (**J**,**K**) mRNA expression analysis of lipolytic and lipogenesis-related genes as in (**H**). Cyclophilin-A was used as internal control. Normalized copies of the indicated genes in resting EU GC muscles were set as 1. Data represent the mean ± SD of nine replicates. * *p* < 0.05, ** *p* < 0.01, *** *p* < 0.001.

**Figure 5 ijms-24-12074-f005:**
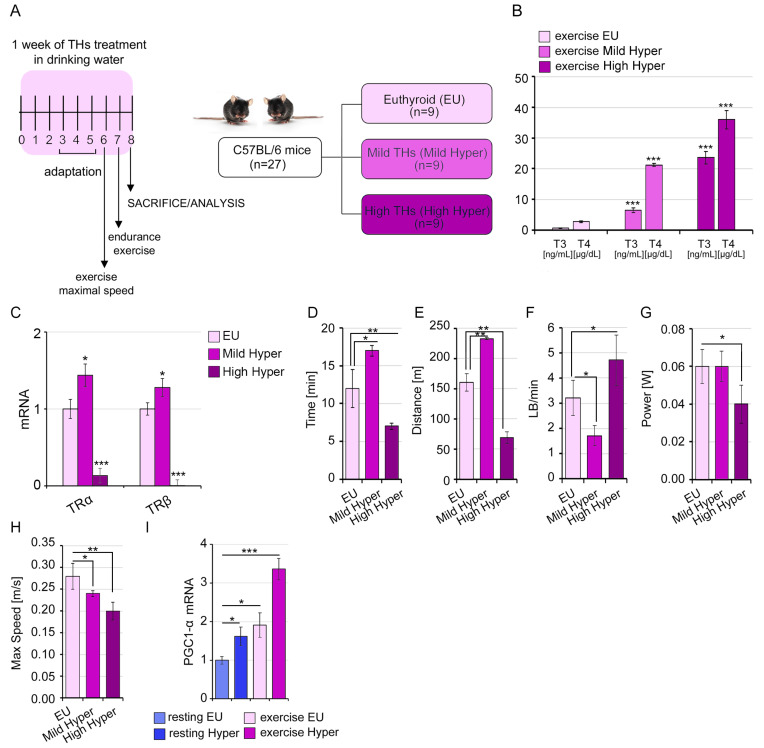
A variable hyperthyroidism grading differentially impacts on SkM exercise performance. (**A**) Schematic representation of experimental setting and exercise-trained animal groups: euthyroid, mild, and high hyperthyroid mice (EU vs. Mild Hyper and High Hyper). (**B**) Serum levels of T3 (ng/mL) and T4 (µg/dL) in exercised-trained euthyroid, mild and high hyperthyroid mice measured by immunoassay. (**C**) mRNA expression levels of TRα and TRβ in exercised Mild Hyper and High Hyper compared to EU GC muscles. Cyclophilin-A was used as internal control. Normalized copies of TR genes in exercised EU GC muscles were set as 1. (**D**) Running time (minutes), (**E**) Distance (meters), (**F**) Number of interruptions per minute (Light beam/min, LB/min), (**G**) Power (Watts) and (**H**) Maximal speed measured in treadmill experiments from euthyroid, mild and high hyperthyroid mice. (**I**) mRNA expression levels of PGC-1α in exercise EU and Mild Hyper GC muscles compared to resting EU and Mild Hyper GC muscles. Cyclophilin-A was used as internal control. Normalized copies of the PGC-1α gene in resting EU GC muscles were set as 1. Data represent the mean ± SD of nine replicates. * *p* < 0.05, ** *p* < 0.01, *** *p* < 0.001.

**Figure 6 ijms-24-12074-f006:**
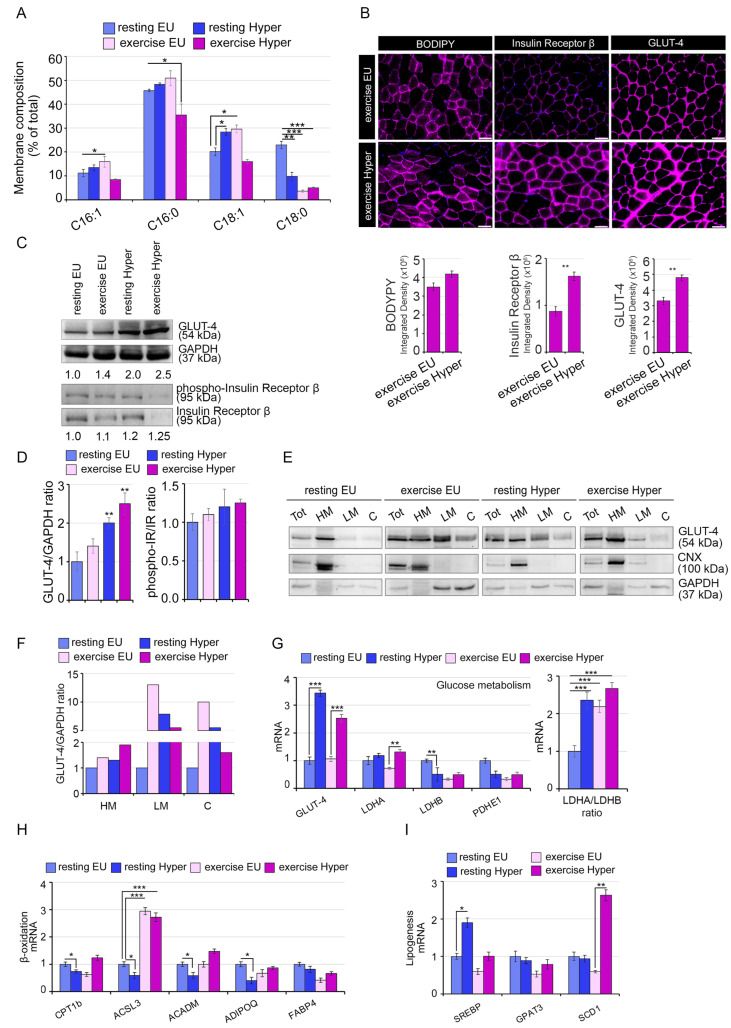
Mild hyperthyroidism alters the structural proprieties of SkM membrane lipid composition. (**A**) Lipidomic analysis performed by GC-MS of SkM membrane lipids extracted from GC muscles of euthyroid and mild hyperthyroid exercise-trained compared to euthyroid and mild hyperthyroid resting mice. Metabolites signals were normalized using internal standards and expressed as Area Under Curve (AUC). Comparisons and differences were analyzed for statistical significance by the two-way Anova test and Bonferroni posttest analysis. * *p* < 0.05, ** *p* < 0.01, *** *p* < 0.001. (**B**) Immunofluorescence analysis of Bodipy for lipid staining, Insulin receptor β and GLUT-4 in TA muscles as in A. Magnification 20×. Scale bar, 50 μm. (**C**) Western blot analysis of GLUT-4 and phosphorylated Insulin receptor β/Insulin receptor β protein in same GC muscles as in A. The numbers below show the quantification of GLUT-4 versus GAPDH levels used as internal loading control and the ratio between the phospho-IR/IR. (**D**) The histograms show the quantification of each antibody versus the loading control. The statistical significance was calculated with respect to resting EU. Data represent the mean ± SD of three gels. ** *p* < 0.01. (**E**) Western blot analysis of distinct fractions: Total homogenate (Tot); Heavy membrane (HM); Light Membrane (LM); Cytosol (**C**) in GC muscles as in C. Calnexin (CNX) levels were used as HM loading control. (**F**) The histograms show the quantification of GLUT-4 versus GAPDH levels used as loading control. (**G**) mRNA expression levels of genes involved in glycolytic pathway in euthyroid and mild hyperthyroid exercise-trained compared to euthyroid and mild hyperthyroid resting GC muscles. Ratio of LDHA/LDHB mRNA expression levels in GC muscle as in (**A**). (**H**,**I**) mRNA expression analysis of lipolytic and lipogenesis-related genes in GC muscle as in (**A**). Cyclophilin-A was used as an internal control. Normalized copies of the indicated genes in resting EU GC muscles were set as 1. Data represent the mean ± SD of nine replicates. * *p* < 0.05, ** *p* < 0.01, *** *p* < 0.001.

**Figure 7 ijms-24-12074-f007:**
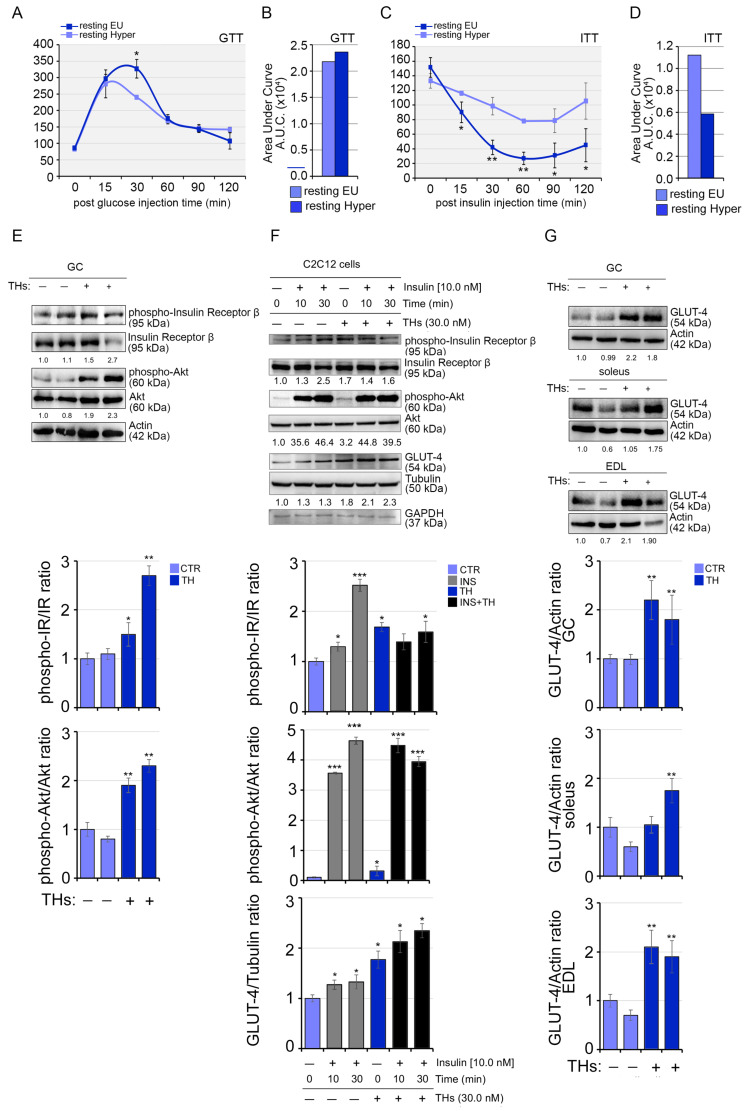
Systemic TH treatment enhances insulin sensitivity in vivo. (**A**,**B**) Glucose tolerance test (GTT) and (**C**,**D**) insulin tolerance test (ITT) were performed in resting hyperthyroid mice versus euthyroid mice. (**E**) Western blot analysis of phospho IR/IR; phospho Akt/Akt and GLUT-4 in GC muscles from resting hyperthyroid versus euthyroid mice after Insulin injection (1 milliUnit of insulin/gm mouse). The histograms below show the quantification of each antibody versus the loading control. The statistical significance was calculated with respect to GC CTR. Data represent the mean ± SD of three gels. * *p* < 0.05, ** *p* < 0.01. (**F**) Western blot analysis of phospho IR/IR; phospho Akt/Akt and GLUT-4 in C2C12 cells treated with THs (30 nM T3/T4) and Insulin (10 nanomolar) alone and in combination at two different time points (10 and 30 min). The histograms below show the quantification of each antibody versus the loading control. The statistical significance was calculated with respect to C2C12 CTR. Data represent the mean ± SD of three gels. * *p* < 0.05, *** *p* < 0.001. (**G**) Western blot analysis of GLUT-4 in GC, soleus and EDL muscles from resting hyperthyroid versus euthyroid mice. The histograms below show the quantification of each antibody versus the loading control. The statistical significance was calculated with respect to GC CTR. Data represent the mean ± SD of three gels. ** *p* < 0.01. The numbers below show the quantification of GLUT-4 versus Tubulin levels used as internal loading control and the ratio between the phospho-IR/IR and phospho Akt/Akt.

**Table 1 ijms-24-12074-t001:** List of oligonucleotides.

Gene	Sense	Sequence
*ACADM*	Forward	AGGGTTTAGTTTTGAGTTGACGG
	Reverse	CCCCGCTTTTGTCATATTCCG
*ACSL3*	Forward	AACCACGTATCTTCAACACCATC
	Reverse	AGTCCGGTTTGGAACTGACAG
*ADIPOQ*	Forward	GTGACGACACCAAAAGGGCTC
	Reverse	TCCAACCTGCACAAGTTCCC
*CPT1b*	Forward	GCACACCAGGCAGTAGCTTT
	Reverse	CAGGAGTTGATTCCAGACAGGTA
*ETFA*	Forward	GCCTCATTGCTCCGTTTTCAG
	Reverse	GCTACTAAGCAGGACACTTCAC
*ETFB*	Forward	CTGTCAAGAGGGTCATCGACT
	Reverse	CACAGAAGGGGTTCATGGAGT
*FABP4*	Forward	TCTCACCTGGAAGACAGCTCC
	Reverse	GCTGATGATCATGTTGGGCTTGG
*GLUT-4*	Forward	CAGAAGGTGATTGAACAGAG
	Reverse	AATGATGCCAATGAGAAA
*GPAT3*	Forward	GTGTCCTAGTGCGCTATTGC
	Reverse	TCTGGGGTCTGTACTGCTTG
*LDHA*	Forward	GGCATGGCTTGTGCCATCAGTATC
	Reverse	GGAGATCCATCATCTCGCCCTTGA
*LDHB*	Forward	GGGAGCTTGTTCCTCCAGAC
	Reverse	TGGGTTGGAAACCACGATGAT
*PDHE1a*	Forward	GAAATGTGACCTTCATCGGCT
	Reverse	TGATCCGCCTTTAGCTCCATC
*PDK3*	Forward	TCCTGGACTTCGGAAGGGATA
	Reverse	GAAGGGCGGTTCAACAAGTTA
*PKM1*	Forward	TATAAGAGGCCTCCACGCTG
	Reverse	GGGCATTGAGATTCCTGCAG
*PKM2*	Forward	TTGGTGAGCACGATAATGGC
	Reverse	GGGCATTGAGATTCCTGCAG
*PYC*	Forward	CTGAAGTTCCAAACAGTTCGAGG
	Reverse	GCAACGAAACACTCGGATG
*SCD1*	Forward	CATTCTCATGGTCCTGCTGC
	Reverse	GCCGTGCCTTGTAAGTTCTG
*SREBP1c*	Forward	ACAGACACTGGCCGAGATG
	Reverse	GCTCTCAGGAGAGTTGGCAC
*TRα*	Forward	ACCACCGCAAACACAACATT
	Reverse	CATTCCGAGAAGCTGCTGTC
*TRβ*	Forward	CACAGGGTACCACTATCGCTGC
	Reverse	CAGCACCAAGTCTGTTGCCATGC

**Table 2 ijms-24-12074-t002:** List of antibodies.

Antibodies Used for Western Blot and Immunofluorescence Analysis			
Antibody	Source	Identifier	Dilution
Insulin Receptor b (4B8)	Cell Signaling Technology (Danvers, MA, USA)	#3025	1:1000 IF
Anti-Glucose Transporter GLUT-4 (1F8)	Abcam (Cambridge, UK)	Ab35826	1:1000 WB/1:500 IF
GAPDH	Elabscience (Houston, TX, USA)	E-AB-20059	1:1000 WB
Mouse monoclonal anti-Myosin Heavy Chain Type I	DSHB (Iowa City, IA, USA)	BA-F8	1:500 IF
Mouse monoclonal anti-Myosin Heavy Chain Type IIA	DSHB	SC-71	1:500 IF
Mouse monoclonal anti-Myosin Heavy Chain Type IIX	DSHB	6H1	1:500 IF
Mouse monoclonal anti-Myosin Heavy Chain Type IIB	DSHB	BF-F3	1:500 IF

## Data Availability

The original contributions presented in the study are included in the article, further inquiries can be directed to the corresponding author.

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
