# Peer review of "Thyroid Hormone Regulates the Lipid Content of Muscle Fibers, Thus Affecting Physical Exercise Performance"

_ijms, 2023, doi:10.3390/ijms241512074_

Round 1

Reviewer 1 Report

ijms-2521429

Major comments

1.     Lines 245, 459: cite publication(s) showing T3 + T4 in drinking water acting as thyroid hormone in rodents

Minor editorial comments

1.     Lines 19, 88, 139: Thyroid Hormones should be Thyroid hormones ?

2.     Lines 81, 317: in vitro, in vivo italic? Follow the journal style

3.     Lines 107, 109, 111: Figures should be Figure ?

4.     Line 122: skeletal muscle cells should be C2C12 skeletal muscle cells

5.     Lines 145, 243: eutyroid should be eu-thyroid ?

6.     Line 152: do not insert line break

7.     Lines 165, 231: add space after 100

8.     Line 174, Fig. 2F, line Lines 214, 234, 293, 294: Insulin Receptor should be Insulin receptor  ?

9.     Line 211: Giacco et al. (2022) and Fortunato et al. (2008)

10. Line 331: minutes should be min

11. Line 354: thyroid receptors should be TRs

12. Line 420: add city, state, country after AL007

13. Line 422: add city, state, country after 25030024

14. Line 425: add city, state, country after respectively

15. Line 430: ad libitum should be italic

16. Line 432: in vivo should be italic ?

17. Line 443: 1 mm EDTA ?

18. Line 447: add space after 30

19. Line 453: measured should be conducted ?

20. Line 468: add city before Germany

21. Line 476: add space after 50

22. Line 492: delete (Kyoto, Japan)

23. Line 502: add space after 4

24. Line 504: add city, state, country after Bayer

25. Line 508: add city, state, country after Lief Technologies

26. Line 511: add Hercules, CA, USA after Bio-Rad

27. Line 525: (Sigma-Aldrich, city, state, country)

28. Lines 529, 535: delete , Hercules, CA  Hercules, CA, USA

29. Line 534: abcam should be Abcam (city, state, country)

30. Line 536: Massachusetts should be MA

31. Line 538: add , Houston, TX, USA after 20059

32. Line 540: Maryland should be MD, USA

33. Lines 553, 555, 568: minutes should be min

34. Line 556: add (city, state, country) after cemtrifuge

35. Line 559: abcam should be Abcam

36. Line 559: abcam should be Abcam    add (city, state, country) after Enzo Life Sciences

37. Line 561: delete , Bethesda, Maryland

38. Line 571: Glycerol should be glycerol

39. Line 567: add city, state after Molecular Probes

40. Line 578: minutes should be min

41. Line 580: mins should be min

42. Line 581: Glycerol should be glycerol

43. Line 583: add (Leica, city, Germany) after Imaging

44. Line 585: Standard Error of the Mean should be standard error of the mean

45. Line 588: cyclophilin-A

46. References: follow the journal style: journal names should be italic; add priod after abbreviated names of journals; titles should not be italic; no need p. for pages; published year should be bold: volume should not be bold; first and last pages such as 205-243; titles should not be large capitals (ref no. 7, 8, 11, 16, 17, 23, 24, 27, 28, 33, 42, 44, 46, 47, 51, 55, 58, 59, 60, 61, 62, 66); pages are missing (ref. no. 7, 8, 17, 24, 44, 59, 60, 62

Author Response

Response to Reviewer 1 Comments

Major comments

Point 1: Lines 245, 459: cite publication(s) showing T3 + T4 in drinking water acting as thyroid hormone in rodents

Response 1: We agree with the reviewer and we have added in text the citations.

Minor editorial comments

Point 1: Lines 19, 88, 139: Thyroid Hormones should be Thyroid hormones?

Response 1: We thank the reviewer; we removed the upper cases from Thyroid Hormones in indicated lines.

Point 2: Lines 81, 317: in vitro, in vivo italic? Follow the journal style

Response 2: We agree with the reviewer and apologize for the mistake. We have used the italic throughout the text for in vitro and in vivo.

Point 3: Lines 107, 109, 111: Figures should be Figure?

Response 3: We have changed Figures in Figure.

Point 4: Line 122: skeletal muscle cells should be C2C12 skeletal muscle cells

Response 4: We agree with the reviewer, we added C2C12 in the sentence.

Point 5: Lines 145, 243: eutyroid should be eu-thyroid?

Response 5: We have changed with eu-thyroid as the reviewer suggests.

Point 6: Line 152: do not insert line break

Response 6: We agree with the reviewer and apologize for the mistake.

Point 7: Lines 165, 231: add space after 100

Response 7: We corrected the mistake.

Point 8: Line 174, Fig. 2F, line Lines 214, 234, 293, 294: Insulin Receptor should be Insulin receptor?

Response 8: We agree with the reviewer and corrected the mistake.

Point 9: Line 211: Giacco et al. (2022) and Fortunato et al. (2008)

Response 9: We thank the reviewer. We have indicated the numbered references.

Point 10: Line 331: minutes should be min

Response 10: We corrected the mistake.

Point 11: Line 354: thyroid receptors should be TRs

Response 11: We agree with the reviewer and corrected the mistake.

Point 12: Line 420: add city, state, country after AL007

Response 12: We added information in the material and methods section.

Point 13: Line 422: add city, state, country after 25030024

Response 13: We added information in the material and methods section.

Point 14: Line 425: add city, state, country after respectively

Response 14: We added information in the material and methods section.

Point 15: Line 430: ad libitum should be italic

Response 15: We agree with the reviewer and apologize for the mistake. We have used the italic throughout the text for ad libitum.

Point 16: Line 432: in vivo should be italic?

Response 16: We agree with the reviewer and apologize for the mistake.

Point 17: Line 443: 1 mm EDTA?

Response 17: We corrected the mistake.

Point 18: Line 447: add space after 30

Response 18: We corrected the mistake.

Point 19: Line 453: measured should be conducted?

Response 19: We agree with the reviewer and substituted “measured” with “conducted”.

Point 20: Line 468: add city before Germany

Response 20: We added information in the sentence.

Point 21: Line 476: add space after 50

Response 21: We added the space.

Point 22: Line 492: delete (Kyoto, Japan)

Response 22:  We deleted (Kyoto, Japan).

Point 23: Line 502: add space after 4

Response 23: We added the space.

Point 24: Line 504: add city, state, country after Bayer

Response 24: We added information in the sentence.

Point 25: Line 508: add city, state, country after Lief Technologies

Response 25: We added information in the sentence.

Point 26: Line 511: add Hercules, CA, USA after Bio-Rad

Response 26: We added information in the sentence.

Point 27: Line 525: (Sigma-Aldrich, city, state, country)

Response 27: We added information in the sentence.

Point 28: Lines 529, 535: delete, Hercules, CA Hercules, CA, USA

Response 28:  We deleted Hercules, CA, USA.

Point 29: Line 534: abcam should be Abcam (city, state, country)

Response 29: We substituted abcam with Abcam and added the information required.

Point 30: Line 536: Massachusetts should be MA

Response 30: We substituted Massachusetts with MA

Point 31: Line 538: add, Houston, TX, USA after 20059

Response 31: We added the information.

Point 32: Line 540: Maryland should be MD, USA

Response 32: We substituted Maryland with MD, USA as required.

Point 33: Lines 553, 555, 568: minutes should be min

Response 33: We have corrected it.

Point 34: Line 556: add (city, state, country) after cemtrifuge

Response 34:  We added information in the sentence.

Point 35: Line 559: abcam should be Abcam

Response 35:  We have corrected it.

Point 36: add (city, state, country) after Enzo Life Sciences

Response 36:  We added information in the sentence.

Point 37: Line 561: delete, Bethesda, Maryland

Response 37:  We deleted it.

Point 38: Line 571: Glycerol should be glycerol

Response 38:  We have corrected it.

Point 39: Line 567: add city, state after Molecular Probes

Response 39:  We added information in the sentence.

Point 40: Line 578: minutes should be min

Response 40:  We have corrected it.

Point 41: Line 580: mins should be min

Response 41:  We have corrected it.

Point 42:  Line 581: Glycerol should be glycerol

Response 42:  We have corrected it.

Point 43: Line 583: add (Leica, city, Germany) after Imaging

Response 43:  We added information in the sentence.

Point 44: Line 585: Standard Error of the Mean should be standard error of the mean 

Response 44: We have corrected it.

Point 45: Line 588: cyclophilin-A

Response 45:  We have corrected it.

Point 46: References: follow the journal style: journal names should be italic; add priod after abbreviated names of journals; titles should not be italic; no need p. for pages; published year should be bold: volume should not be bold; first and last pages such as 205-243; titles should not be large capitals (ref no. 7, 8, 11, 16, 17, 23, 24, 27, 28, 33, 42, 44, 46, 47, 51, 55, 58, 59, 60, 61, 62, 66); pages are missing (ref. no. 7, 8, 17, 24, 44, 59, 60, 62

Response 46:  We apologize for the incorrect reference style and missing information. We have now re-edited the references section following the journal style.

Additional comments:

  1. Please mention Table 1 and 2 in the text.

We have mentioned table in the text

  1. We have noticed that you insert figure as Table 1 and 2. Please insert table and type all information of Table 1 and 2.

We have inserted Table 1 and 2 as text.

Reviewer 2 Report

ID: ijms-2521429

Thyroid Hormone Regulates the Lipid Content of Muscle Fibers, Thus Affecting Physical Exercise Performance. by Miro, C et al.

To the Authors:

General comments:

The authors investigated the effects of thyroid hormones (THs) on the lipid metabolism and function of skeletal muscle (SkM).  They found that THs induced a remodeling of lipid composition: THs reduced the ratio of stearic/oleic acid in SkM.  Also, it was demonstrated that the saturation index of the cell membrane was associated with an improvement of insulin sensitivity and endurance exercise.  It was considered that this study was structured well and includes novelty; however, several points should be addressed to improve the manuscript.

Specific comments:

1. The rationale to reproduce various hyperthyroid conditions by using T3 and T4, in vivo and in vitro, should be addressed in detail in the results or introduction sections.

2. Measuring intramuscular THs levels in exercised-trained euthyroid, mild and high hyperthyroid mice would be useful for understanding exact effects of THs on SkM exercise performance (regarding Fig. 5). 

3. The expression of TH receptors in each in vivo and in vitro experiment are needed and, if possible, the expressional changes of TH receptors in the different hyperthyroid status would be informative for the present concept.

4. The authors should add the quantitative analysis in western blotting (regarding Fig. 6C-D and 7E-G).

Author Response

Response to Reviewer 2 Comments

The authors investigated the effects of thyroid hormones (THs) on the lipid metabolism and function of skeletal muscle (SkM). They found that THs induced a remodeling of lipid composition: THs reduced the ratio of stearic/oleic acid in SkM.  Also, it was demonstrated that the saturation index of the cell membrane was associated with an improvement of insulin sensitivity and endurance exercise.  It was considered that this study was structured well and includes novelty; however, several points should be addressed to improve the manuscript.

Specific comments:

Point 1: The rationale to reproduce various hyperthyroid conditions by using T3 and T4, in vivo and in vitro, should be addressed in detail in the results or introduction sections.

Response 1:  We thank the reviewer for rising this point. Regarding this issue, it is well established that circulating and tissue THs levels are maintained in a relatively narrow concentration range and that alterations in blood TH levels can have detrimental effects in a variety of tissues, including heart, kidney, liver, and brain. In light of these considerations, in in vitro experiments, we choose a low dosage treatment (3nM T3 and T4) to avoid the effects of a supraphysiological exposure, which is revealed effective to mediate the biological effects on lipid membrane composition, insulin sensitivity and gene expression. Accordingly, and also, to prevent the side effects of the induced systemic hyperthyroidism, we tested two different doses of T3 and T4 in in vivo settings, referred as mild and high hyperthyroidism, to evaluate changes in circulating TH levels, as well as downstream effects on physiological and muscle parameters, and also gene expression. A more detailed explanation has been included in the revised version of the manuscript (Results: Lines 95 and 255).

Point 2: Measuring intramuscular THs levels in exercised-trained euthyroid, mild and high hyperthyroid mice would be useful for understanding exact effects of THs on SkM exercise performance (regarding Fig. 5).

Response 2:  We thank the reviewer for the suggestion. We have measured the intramuscular THs levels in exercised muscles and the results are shown in figure below (see attached file). We did not appreciate significant differences in intramuscular T3 and T4 levels in mild hyperthyroidism condition, while a slight decline was observed in high hyperthyroidism, probably suggesting a compensatory peripheral mechanism following one week of TH exposure.

Point 3: The expression of TH receptors in each in vivo and in vitro experiment are needed and, if possible, the expressional changes of TH receptors in the different hyperthyroid status would be informative for the present concept.

Response 3: We thank the reviewer for rising this point. We have measured TH receptors expression in in vitro and in vivo settings. Regarding to the in vitro experiment, the C2C12 cells expressed the TRa isoform, while TRb was only barely detected in our experimental conditions. We have observed that THs treatment induced a slight increase of TRa expression.

The analysis of TRa and TRb expression in GC muscles have shown that the mild hyperthyroidism was associated with an increase of TRa and TRb expression in both resting and exercised muscles, while the high dose treatment was associated with a reduction of TH receptors. Importantly, the exercise stimulus in euthyroid mice induced an increase of TR receptors expression. However, when exercised mice were exposed to the higher TH dosage, the TR expression was suppressed. This data may suggest that skeletal muscles have compensatory mechanisms to prevent tissue-level alterations, including downregulation of receptors, in response to strong systemic hyperthyroidism. These results also further confirm that the high THs exposure induces an alteration of the normal physiological THs signaling.

The observation that a mild THs treatment elevates the expression of TH receptors further reinforces the concept (and one of the main messages of our work) that a slight elevation of THs signal might be beneficial to the overall exercise performance. The new data are inserted in Fig 1I, Fig 2D, Fig 4E, and Fig 5D and commented in the Results (paragraph 2.1, 2.2, 2.3 and 2.4) and Discussion section (pag 15) of the revised version of the manuscript.

Point 4: The authors should add the quantitative analysis in western blotting (regarding Fig. 6C-D and 7E-G).

Response 4: We apologize if the results are not clearly appreciable. For gel blots in Fig 6C the quantification is indicated by numbers below showing the quantification of the protein of interest versus the internal loading control. For gel blots in Fig 6D the quantification is indicated in the histogram in Fig 6E. For the gel blots in Fig 7E-G the quantification is indicated by numbers below showing the quantification of the protein of interest versus the internal loading control.

Additional comments:

  1. Please mention Table 1 and 2 in the text.

We have mentioned table in the text

  1. We have noticed that you insert figure as Table 1 and 2. Please insert table and type all information of Table 1 and 2.

We have inserted Table 1 and 2 as text.

Round 2

Reviewer 2 Report

ID: ijms-2521429

Thyroid Hormone Regulates the Lipid Content of Muscle Fibers, Thus Affecting Physical Exercise Performance. by Miro, C et al.

To the Authors:

General comments:

It is considered that the authors successfully revised the manuscript according to the comments.  As for the quantitative analysis for western blots shown in Fig. 6 and 7, it would be better to perform the statistical analysis.

Author Response

Response to Reviewer 2 Comments

General comments: It is considered that the authors successfully revised the manuscript according to the comments.  As for the quantitative analysis for western blots shown in Fig. 6 and 7, it would be better to perform the statistical analysis.

Response 1:  We thank the reviewers for the suggestion. We have performed statistical analysis of the gels run in triplicate in figures 6 and 7 and inserted the histograms illustrating the quantifications and the statistical significance in the relative figures.
